# Amide Proton Transfer–Chemical Exchange Saturation Transfer Imaging of Intracranial Brain Tumors and Tumor-like Lesions: Our Experience and a Review

**DOI:** 10.3390/diagnostics13050914

**Published:** 2023-02-28

**Authors:** Hirofumi Koike, Minoru Morikawa, Hideki Ishimaru, Reiko Ideguchi, Masataka Uetani, Mitsuharu Miyoshi

**Affiliations:** 1Department of Radiology, Nagasaki University Graduate School of Biomedical Sciences, 1-7-1 Sakamoto, Nagasaki 852-8501, Japan; 2Department of Radiology, Nagasaki University Hospital, 1-7-1 Sakamoto, Nagasaki 852-8501, Japan; 3Department of Radioisotope Medicine, Nagasaki University Graduate School of Biomedical Sciences, 1-7-1 Sakamoto, Nagasaki 852-8588, Japan; 4MR Application and Workflow, GE Healthcare Japan, Hino 191-0065, Japan

**Keywords:** magnetic resonance imaging, amide proton transfer imaging, brain tumor, tumor like lesion

## Abstract

Chemical exchange saturation transfer (CEST) is a molecular magnetic resonance imaging (MRI) method that can generate image contrast based on the proton exchange between labeled protons in solutes and free, bulk water protons. Amide proton transfer (APT) imaging is the most frequently reported amide-proton-based CEST technique. It generates image contrast by reflecting the associations of mobile proteins and peptides resonating at 3.5 ppm downfield from water. Although the origin of the APT signal intensity in tumors is unclear, previous studies have suggested that the APT signal intensity is increased in brain tumors due to the increased mobile protein concentrations in malignant cells in association with an increased cellularity. High-grade tumors, which demonstrate a higher proliferation than low-grade tumors, have higher densities and numbers of cells (and higher concentrations of intracellular proteins and peptides) than low-grade tumors. APT-CEST imaging studies suggest that the APT-CEST signal intensity can be used to help differentiate between benign and malignant tumors and high-grade gliomas and low-grade gliomas as well as estimate the nature of lesions. In this review, we summarize the current applications and findings of the APT-CEST imaging of various brain tumors and tumor-like lesions. We report that APT-CEST imaging can provide additional information on intracranial brain tumors and tumor-like lesions compared to the information provided by conventional MRI methods, and that it can help indicate the nature of lesions, differentiate between benign and malignant lesions, and determine therapeutic effects. Future research could initiate or improve the lesion-specific clinical applicability of APT-CEST imaging for meningioma embolization, lipoma, leukoencephalopathy, tuberous sclerosis complex, progressive multifocal leukoencephalopathy, and hippocampal sclerosis.

## 1. Introduction

Chemical exchange saturation transfer (CEST) is an important molecular magnetic resonance imaging (MRI) technique that can generate image contrast based on the proton exchange between labeled protons in solutes and free, bulk water protons [1,2]. Amide proton transfer (APT) imaging is the most frequently reported amide-proton-based CEST technique. It generates image contrast by reflecting the associations of mobile proteins and peptides resonating at 3.5 ppm downfield from water [3,4]. APT-CEST does not require an exogenous contrast and is safe in patients with renal failure or with an intolerance to contrast media. The technique has shown a potential clinical utility for differentiating radiation necrosis from tumor recurrence or progression and for differentiating between high-grade and low-grade glioma [5,6,7]. Although the origin of the APT signal intensity (SI) in tumors is unclear, previous studies have suggested that the APT SI is increased in brain tumors because of increased mobile protein concentrations in malignant cells in association with increased cellularity [3,6,7,8,9,10,11]. Previous reports reached a consensus view that the APT SI is higher in highly malignant tumors than in less-malignant tumors and is also higher in malignant tumors than in non-tumor lesions such as radiation necrosis. However, there are few reports on the APT SI of benign tumors and tumor-like lesions including demyelinating disease. We believe that the APT-CEST technique is suitable for application in a wider ranging field, including benign tumors and tumor-like lesions such as demyelinating disease. Therefore, our aim was to review the current applications and findings of the APT-CEST imaging of various brain tumors and tumor-like lesions to help differentiate between benign and malignant tumors and to estimate the nature of lesions.

## 2. Mechanism of APT CEST Imaging

The APT effect is measured as a reduction in bulk water intensity due to the chemical exchange of water protons with labeled backbone amide protons of endogenous mobile proteins and peptides in tissue [3,4]. Thus, specific molecular information is obtained indirectly through the bulk water signal. Such labeling is accomplished using selective radiofrequency (RF) irradiation at the MR frequency of the backbone amide protons, which are 3.5 ppm downfield of the water resonance, causing saturation that is transferred to water protons (Figure 1).

## 3. MRI Protocol

All patients were examined with a 3-T MRI system (Signa™ Architect, GE Healthcare, Milwaukee, WI, USA) equipped with a 48 channel receiver array coil. APT-CEST images were acquired before contrast administration using single-shot, fast spin echo sequences with the following parameters: field of view = 220 × 220 mm; matrix = 128 × 128; spatial resolution = 1.7 × 1.7 mm; slice thickness = 8.0 mm; repetition time/echo time (TR/TE) = 3000/26.6 ms; and number of slices = 1.

Twenty-nine saturation frequency offsets from −7.0 to 7.0 ppm in increments of 0.5 ppm were used to attain a sufficient signal-to-noise ratio within the clinical time frame.

The APT imaging consisted of RF saturation (one pulse with a duration of 2000 ms) and an average B_1_ radiofrequency field equivalent to a continuous RF power level of 2.0 µT (microtesla).

The water frequency shift due to field inhomogeneity was measured in a separate image acquired using the water saturation shift referencing method with 11 offset frequencies ranging from −1.875 to 1.875 ppm at intervals of 0.375 ppm. One reference image was acquired without a saturation RF pulse, resulting in a full Z-spectrum within the offset range. The water saturation shift reference image was acquired with a TR/TE of 3000/26.6 ms, an RF saturation amplitude of 0.5 µT, and a total duration of 2000 ms with a continuous wave. The total acquisition time for both the APT and water-saturation shift reference images was 2 min 9 s.

## 4. APT-CEST Image Processing

The APT data were analyzed in MATLAB (MathWorks, Inc., Natick, MA, USA). The magnetization transfer ratio asymmetry (*MTR_asym_*) was also calculated. According to the shift-corrected data, the *MTR_asym_* values at ±3.5 ppm with respect to water frequency were calculated as follows [12,13,14]:$$MTR_{asym\;}\left(+3.5_{ppm}\right)=\frac{S_{sat}\left(-3.5_{ppm}\right)-S_{sat}\left(+3.5_{ppm}\right)}{S_{0}},$$
where *S_sat_* is the SI with selective imaging and *S*_0_ is the SI in the absence of RF for imaging SI normalization. B0 inhomogeneity was corrected on a pixel-by-pixel basis with a water saturation reference map.

## 5. APT-CEST Imaging of Malignant Tumor

### 5.1. Oligodendroglioma

Oligodendroglial tumors are diffuse, infiltrating gliomas that most commonly arise in the frontal lobe. They have a peak incidence in the fourth or fifth decade of life. They are the second most common intracranial glial tumors and are mainly composed of World Health Organization (WHO) grade II and III oligodendrogliomas [15,16] (Figure 2). 

### 5.2. Diffuse Astrocytoma

The 2016 WHO classification of diffuse astrocytic tumors indicates three grades with different aggressiveness [17]. Although diffuse astrocytoma (WHO grade II) is a relatively slow-growing tumor with a median survival time of 5–8 years, it has a high recurrence rate due to its diffuse infiltration of brain tissue and inherent malignant potential to transform into a high-grade astrocytoma such as anaplastic astrocytoma or secondary glioblastoma [18,19] (Figure 3).

### 5.3. Glioblastoma

Glioblastoma (GBM) is the most common malignant brain tumor in adults. With a designation of WHO Grade IV, it is also the most lethal primary brain malignancy, with a median survival time of only 15 months [20,21] (Figure 4 and Figure 5).

APT-CEST imaging shows an iso-intensity or mild punctate hyperintensity in low-grade gliomas which is significantly lower than what is seen in high-grade gliomas [7]. There are also significant differences in APT SI between grade II and III glioma and grade III and IV glioma [6]. In our case, the APT SI was higher in GBM than in oligodendroglioma and diffuse astrocytoma.

### 5.4. Malignant Lymphoma

Intracranial lymphoma may present as essentially primary central nervous system (CNS) B-cell non-Hodgkin lymphoma (B-cell PCNSL), metastatic CNS lymphoma, intravascular lymphomatosis, T-cell PCNSL, and intracranial Hodgkin’s lymphoma [22,23]. Primary central nervous system lymphoma is responsible for 1–2% of all central nervous system tumors. It encompasses lymphoma exclusively involving the brain, spinal cord, eyes, meninges, and cranial nerves [24].

Primary central nervous system lymphomas show more homogeneous APT SI than high-grade gliomas. The maximum APT SI in primary central nervous system lymphomas was lower than in high-grade gliomas [25]. In our case, B cell lymphoma showed a high APT SI (Figure 6).

### 5.5. Brain Metastasis

Brain metastases (BMs) are associated with a poor prognosis irrespective of the primary tumor they originate from. Lung cancer, breast cancer, and melanoma are the most common causes of BM, accounting for 67–80% of cases [26]. A previous report showed that APT SIs in perilesional tissue in GBM were significantly lower in the solitary brain metastases [27]. Moreover, one report showed that the APT SI of solitary BMs was lower than that of enhanced areas of GBM [28]. In our case, the margin of BMs showed a relatively high APT SI (Figure 7).

## 6. APT-CEST Imaging of Meningioma

Meningioma is the most frequently diagnosed primary brain tumor in adults [29,30]. Meningiomas are categorized into three WHO grades with 15 histological subtypes, indicating heterogenous clinical and molecular genetic characteristics [31]. As most meningiomas are benign and categorized as WHO Grade I with a slow-growing behavior, most require no immediate treatment. However, some subtypes corresponding to WHO grades II and III are associated with a higher risk of recurrence and shorter survival times. One study investigated the feasibility of APT-CEST imaging for differentiating benign from atypical meningiomas [32] (Figure 8), while another study investigated the feasibility of APT-CEST imaging for differentiating growing meningiomas from non-growing meningiomas [33] (Figure 9, Figure 10 and Figure 11).

Preoperative transcatheter arterial embolization (TAE) of meningiomas with polyvinyl alcohol microparticles has often been performed and is considered a safe, efficient, and cost-effective method with few complications [34,35]. However, to our knowledge, there are no reports of APT-CEST imaging in connection with the embolization of a meningioma. In our case, the APT SI of an atypical meningioma decreased after TAE (Figure 8).

## 7. APT-CEST Imaging of Benign Lesion

### 7.1. Sphenoid Sinus Mucocele

Paranasal sinus mucocele is defined as the accumulation and retention of mucoid secretion within the sinus, leading to the thinning, distension, and erosion of its bony walls. Sphenoid sinus mucocele is pathologically benign and comprises 1–2% of all paranasal sinus mucoceles [36]. To the best of our knowledge, there are no reports of APT-CEST imaging of sphenoid sinus mucocele. However, this lesion is not a tumor, and a low APT SI is expected. In our case, sphenoid sinus mucocele showed a low APT SI (Figure 12).

### 7.2. Solitary Fibrous Tumor

The solitary fibrous tumor (SFT) is a new combined entity for grade I–III soft-tissue tumors. It was introduced in the 2016 World Health Organization classification of tumors of the central nervous system [17]. In our case, this tumor was pathologically diagnosed as grade I after surgery. There are no reports on APT-CEST imaging of SFT. However, this lesion is a benign tumor, and we expected that its APT SI would be lower than that of a malignant tumor because SFT typically shows low proliferation. In our case, SFT showed a low APT SI (Figure 13).

### 7.3. Schwannoma

Schwannoma is a benign tumor and develops from the Schwann sheath.

The cranial nerve that is most often affected (in 90% of cases) is the vestibulocochlear nerve (cranial nerve VIII), followed by the trigeminal nerve (cranial nerve V). Acoustic schwannomas grow into the cerebellopontine angle, displacing the brainstem and cerebellum. In most cases, these originate from within the internal auditory canal, the dilation of which is an early radiological sign of tumor growth [37].

This lesion is a benign tumor, and we expected that the APT SI would be lower than that of malignant tumor because the proliferation ability of schwannoma is expected to be low. However, schwannoma showed a high APT SI in our case (Figure 14). A recent study showed that the APT SI in schwannomas with high SI on a T2-weighted sequence was higher than that of meningiomas [38].

### 7.4. Lipoma

Intracranial lipomas are rare growths that represent less than 0.1% of all brain tumors. Most intracranial lipomas are asymptomatic and are found incidentally on imaging while assessing other conditions [39,40,41,42]. There are no reports on the APT-CEST imaging of a lipoma. This lesion is a benign tumor, and we expected the APT SI of intracranial lipoma would be lower than that of malignant tumors because the proliferation ability of lipomas is expected to be low. In our case, the suspected lipoma showed a low APT SI (Figure 15).

## 8. APT-CEST Imaging of Demyelinating Disease and Tumor-like Lesion

### 8.1. Radiation Necrosis

The occurrence and extent of radiation necrosis depends on the age of the patient receiving radiotherapy and the lesion volume [43]. A previous study reported that the APT SI of a glioma was higher than that of radiation necrosis [5]. In our case, radiation necrosis showed a low APT SI (Figure 16 and Figure 17).

### 8.2. Leukoencephalopathy

Toxic leukoencephalopathy is characterized by progressive damage to the white matter. Its causes include a wide spectrum of diseases that may injure and cause structural alteration to the white matter. The insults may be due to toxic metabolites secondary to chemotherapy or immunosuppressive therapy, environmental, or infectious in origin [44]. There are no reports on APT-CEST imaging of leukoencephalopathy. However, this lesion is not a tumor, and we expected the APT SI to be low because the proliferation ability of leukoencephalopathy is expected to be low. In our case, suspected leukoencephalopathy showed a low APT SI (Figure 18).

### 8.3. Tuberous Sclerosis Complex

Tuberous sclerosis complex (TSC) is an autosomal dominant disorder with high clinical variability. It shows various features on brain imaging, including subependymal nodules, radial bands, cortical hamartomas, and subependymal giant cell astrocytomas [45].

In our case, this lesion was suspected to be cortical hamartoma. There are no reports on APT-CEST imaging of TSC and hamartoma. The lesions are benign, and we expected that the APT SI would be lower than that of malignant tumor because the proliferation ability of hamartoma is expected to be low. In our case, the suspected hamartoma showed a low APT SI (Figure 19).

### 8.4. Progressive Multifocal Leukoencephalopathy

Progressive multifocal leukoencephalopathy (PML) is a rare but often fatal brain disease caused by the reactivation of the polyomavirus JC (JCV) [46]. PML almost exclusively affects immunocompromised patients such as those with HIV/AIDS [47].

In systemic lupus erythematosus, risk factors for CNS infection include disease activity, current and cumulative corticosteroid dose, and the use of other immunosuppressants [48,49] 

There are no reports on APT-CEST imaging of PML. However, this lesion is not a tumor, and we expected the APT SI would be low because the proliferation ability of PML is expected to be low. In our case, PML showed a low APT SI (Figure 20).

### 8.5. Hippocampal Sclerosis

Hippocampal sclerosis (HS) is the most common histopathologic abnormality found in adults with drug-resistant temporal lobe epilepsy [50]. There are no reports on APT-CEST imaging of HS. However, this lesion is not a tumor, and we expected the APT SI to be low because the proliferation ability of HS is expected to be low. In our case, PML showed a low APT SI (Figure 21).

## 9. Discussion

Recent studies on gliomas have demonstrated a positive correlation of hte APT SI with the cell proliferation index [6,8,11]. These results suggest that high-grade tumors, which demonstrate a higher proliferation, have higher densities and numbers of cells (and higher concentrations of intracellular proteins and peptides) than low-grade tumors.

The concentration of mobile proteins and peptides per cell may increase with the grade of glioma. In the study of Togao et al., tumors with necrosis showed a higher APT SI than those without necrosis. Although it was difficult to confirm whether this was a direct relationship, highly concentrated mobile proteins and peptides in the extracellular space, such as microscopic necrosis or fluid collection in the microcystic space, might also increase the APT SI in tumors. The alternation in tissue pH might affectthe APT SI. [1,4,6]

Our case also showed that high-grade tumors tended to have a higher APT SI than low-grade tumors, and that demyelinating disease and tumor-like lesions tend to have a low APT SI.

TAE is a standard pre-operative procedure for meningiomas aimed at reducing intra-operative bleeding and facilitating dissection. In our case of atypical meningioma after TAE, contrast-enhanced, T1-weighted sequences showed no apparent change to the meningioma, but APT-CEST imaging showed a decrease in the SI (Figure 8). After the operation in our case, a part of the tumor showed necrosis and ischemic change upon pathological analysis.

APT-CEST imaging may be used as an indicator of therapeutic effects on tumors. Moreover, APT-CEST imaging does not involve the injection of a contrast agent and can be used in patients with renal failure and those who show adverse reactions to contrast media.

There have been few reports on APT-CEST imaging of benign lesions. In our case, they tended to show low APT SIs. However, schwannoma (Figure 14) showed a high APT SI consistent with T2 hyperintensity. The area showing T2 hyperintensity suggests a cystic or heterogeneous appearance. Small schwannomas are usually homogeneous and are histologically composed of Antoni type A pattern, while heterogeneous and cystic schwannomas are larger and include Antoni B pattern or a mix of type A and B patterns [51]. Therefore, this area may show higher proliferation and a high APT SI, and one study showed a higher APT SI in schwannomas with a high SI on a T2-weighted sequence compared to that of meningiomas [38]. Zhang et al. indicated that APT imaging may have some value in the determination of malignant brain tumor boundaries, but there is no doubt that this approach can be used in the differential diagnosis between gliomas and meningiomas [52].

Similarly, the non-contrast region of GBM (Figure 4) also showed a high APT SI, suggesting that a liquid containing high concentrations of intracellular proteins and peptides. However, sinus mucocele (Figure 12) shows low a APT SI because although this lesion may contain viscous fluid, there are less intracellular proteins and peptides. However, the number of cases examined was small, and further studies are needed. In addition, because the evaluations described in this study were made based on a visual assessment, quantitative evaluation is also necessary.

## 10. Conclusions

APT-CEST imaging can provide additional information on intracranial brain tumors and tumor-like lesions over the information provided by conventional MRI methods. This which may help indicate the nature of lesions, differentiate between benign and malignant lesions, and determine therapeutic effects. Future research could initiate or improve the lesion-specific clinical applicability of APT-CEST imaging for meningioma embolization, lipoma, leukoencephalopathy, TSC, PML, and HS. For these purposes, we may have to answer specific research questions and suggest guidelines to allow the development of APT-CEST imaging to reach its full potential.

## Figures and Tables

**Figure 1 diagnostics-13-00914-f001:**
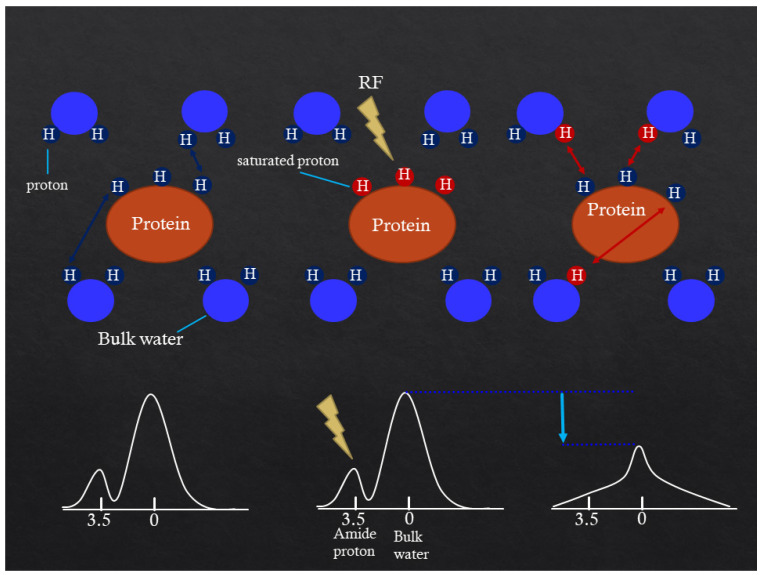
Mechanism of APT CEST imaging. The APT effect is measured as a reduction in bulk water intensity due to chemical exchange of water protons with labeled backbone amide protons at 3.5 ppm.

**Figure 2 diagnostics-13-00914-f002:**
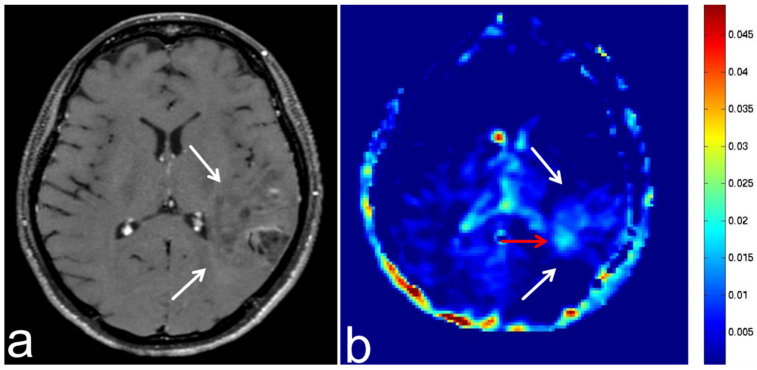
Images from a 35-year-old woman with an oligodendroglioma. (**a**) Three-dimensional, contrast-enhanced, T1-weighted sequence shows ill definition and faint enhancement of an oligodendroglioma in the left temporal lobe (white arrow). (**b**) APT-CEST sequence (axial section) shows wide low SI (white arrow) and parts with relatively high SI (red arrow), consistent with oligodendroglioma.

**Figure 3 diagnostics-13-00914-f003:**
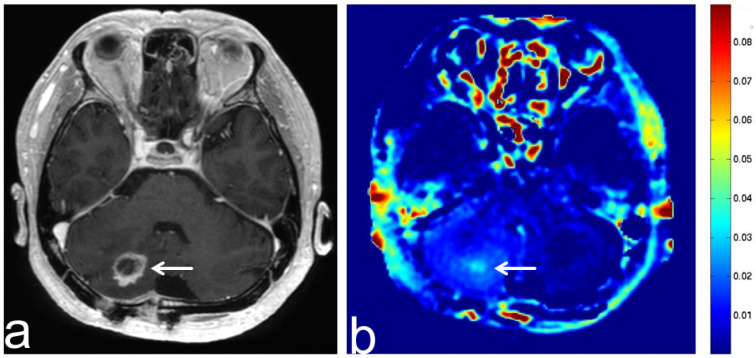
Images from a 41-year-old man with diffuse astrocytoma post-operation. (**a**) Three-dimensional, contrast-enhanced, T1-weighted sequence shows ring enhancement in the right cerebellar hemisphere (white arrow). Local recurrence or malignant transformation is suspected. (**b**) APT-CEST sequence (axial section) shows relatively high SI in the center of a ring of enhancement (white arrow).

**Figure 4 diagnostics-13-00914-f004:**
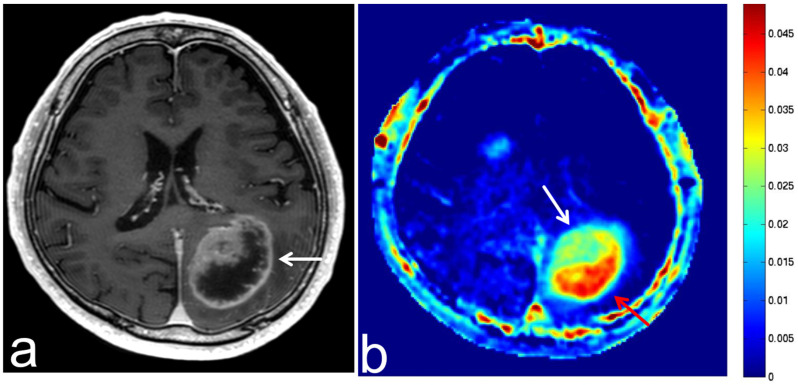
Images from a 77-year-old man with a GBM. (**a**) Three-dimensional, contrast-enhanced, T1-weighted sequence shows ring enhancement of a GBM in the left deep periventricular white matter and adjacent splenium of corpus callosum (white arrow). (**b**) APT-CEST sequence (axial section) shows high SI consistent with GBM (white arrow) and extremely high SI consistent with a poor contrast area (red arrow).

**Figure 5 diagnostics-13-00914-f005:**
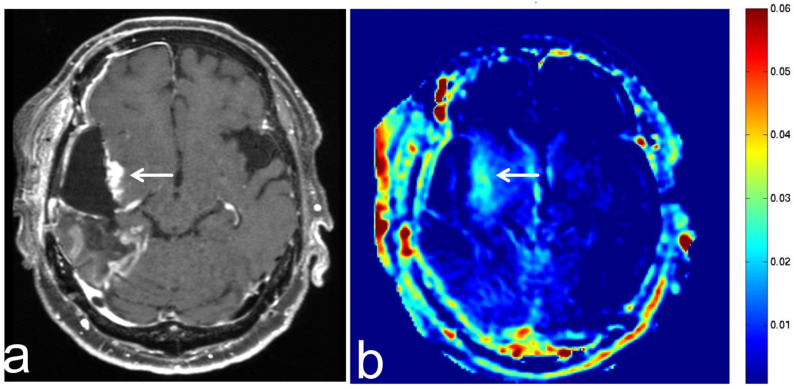
Images from a 72-year-old woman with a GBM after operation and radiotherapy. (**a**) Three-dimensional, contrast-enhanced, T1-weighted sequence shows enhancement at the marginal dead space of the surgical site in the right temporal lobe (white arrow). Local recurrence is suspected according to the clinical course. (**b**) APT-CEST sequence (axial section) shows relatively high SI consistent with the enhancement (white arrow).

**Figure 6 diagnostics-13-00914-f006:**
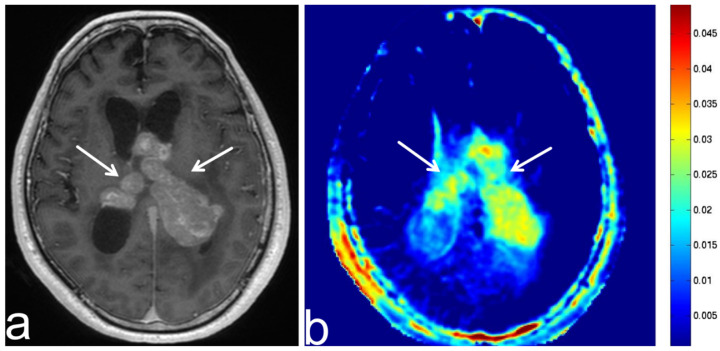
Images from a 68-year-old man with diffuse large B cell lymphoma. (**a**) Three-dimensional, contrast-enhanced, T1-weighted sequence shows an intense enhancement intraventricular mass (white arrow). (**b**) APT-CEST sequence (axial section) shows a high SI throughout the intraventricular malignant lymphoma (white arrow).

**Figure 7 diagnostics-13-00914-f007:**
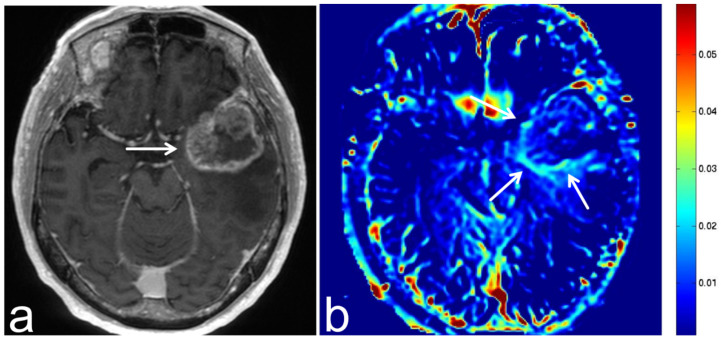
Images from a 70-year-old man with brain metastasis from esophageal cancer. (**a**) Three-dimensional, contrast-enhanced, T1-weighted sequence shows ring enhancement of brain metastasis in the left temporal lobe (white arrow). (**b**) APT-CEST sequence (axial section) shows relatively high SI at the margin of the brain metastasis (white arrow).

**Figure 8 diagnostics-13-00914-f008:**
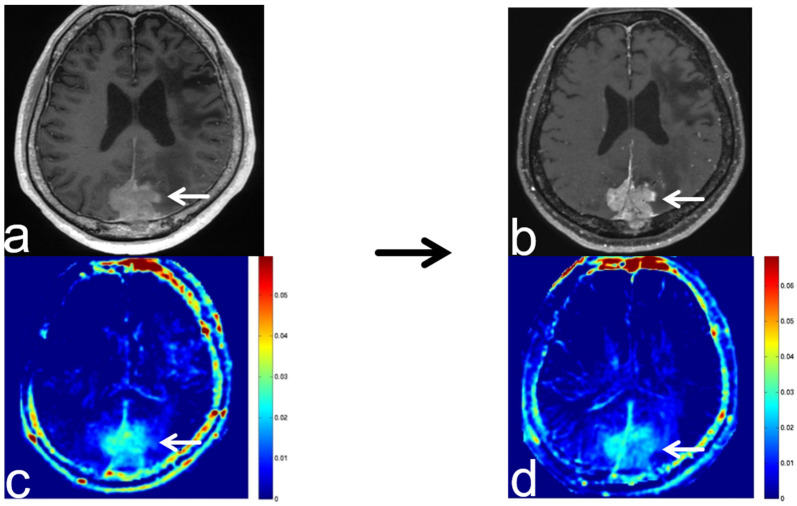
Images from a 64-year-old woman with an atypical meningioma before and after transcatheter arterial embolization (TAE). (**a**) Three-dimensional, contrast-enhanced, T1-weighted sequence shows atypical falcine meningioma involving superior sagittal sinus before TAE (white arrow). (**b**) Three-dimensional, contrast-enhanced, T1-weighted sequence shows atypical falcine meningioma involving superior sagittal sinus after TAE (white arrow). There was no apparent change in the atypical meningioma compared with pre-TAE imaging. (**c**) APT-CEST sequence (axial section) shows high SI consistent with meningioma before TAE (white arrow). (**d**) APT-CEST sequence (axial section) shows a decrease in SI in the left side of the atypical meningioma after TAE (white arrow). However, color bar and range for (**c**,**d**) were slightly different.

**Figure 9 diagnostics-13-00914-f009:**
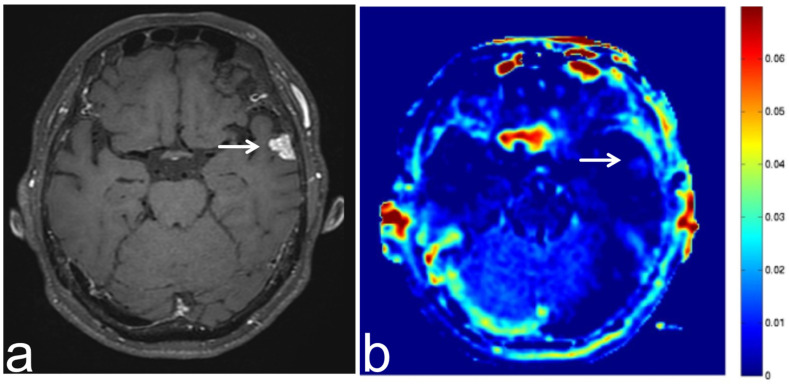
Images from a 74-year-old woman with a non-growing intracranial meningioma. (**a**) Three-dimensional, contrast-enhanced, T1-weighted sequence shows a convexity meningioma in the left temporal lobe (white arrow). (**b**) APT-CEST sequence (axial section) shows low SI consistent with meningioma (white arrow).

**Figure 10 diagnostics-13-00914-f010:**
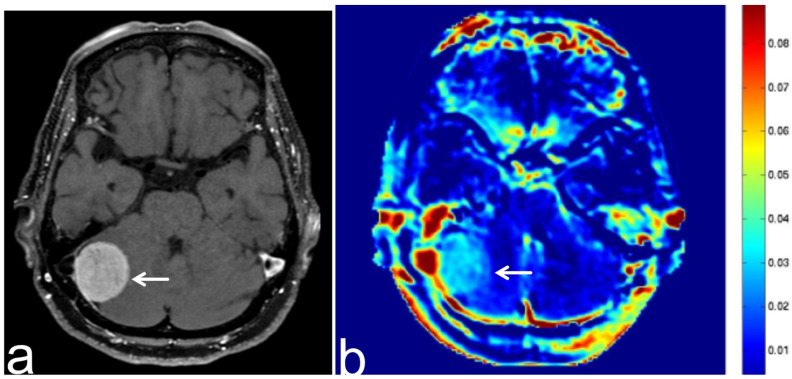
Images from a 77-year-old woman with a growing intracranial meningioma. (**a**) Three-dimensional, contrast-enhanced, T1-weighted sequence shows a dural attached lesion in the right posterior fossa (white arrow). (**b**) APT-CEST sequence (axial section) shows relatively high SI consistent with meningioma (white arrow).

**Figure 11 diagnostics-13-00914-f011:**
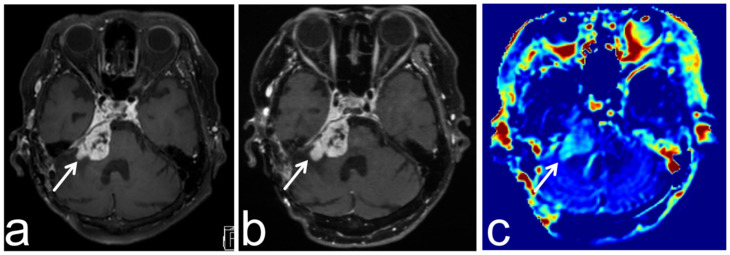
Images from a 67-year-old woman with a growing meningioma post operation. (**a**,**b**) Three-dimensional, contrast-enhanced, T1-weighted sequence shows partially increasing cerebellopontine angle meningioma (white arrow). (**c**) APT-CEST sequence (axial section) shows relatively high SI consistent with the increasing part of the tumor (white arrow).

**Figure 12 diagnostics-13-00914-f012:**
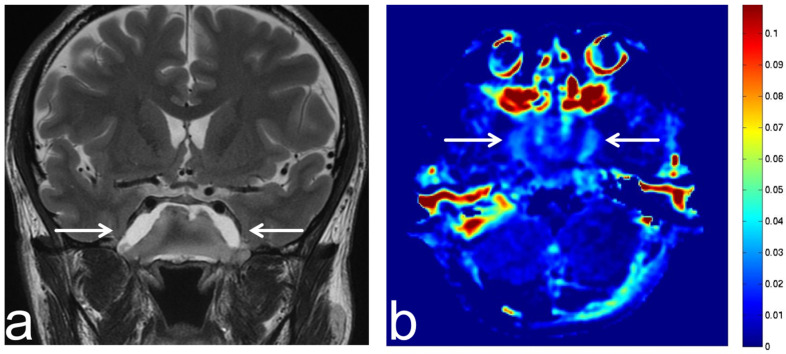
Images from a 38-year-old man with sphenoid sinus mucocele. (**a**) T2-weighted sequence (coronal section) shows a sphenoid sinus mucocele (white arrow). (**b**) APT-CEST sequence (axial section) shows low SI consistent with sphenoid sinus mucocele (white arrow).

**Figure 13 diagnostics-13-00914-f013:**
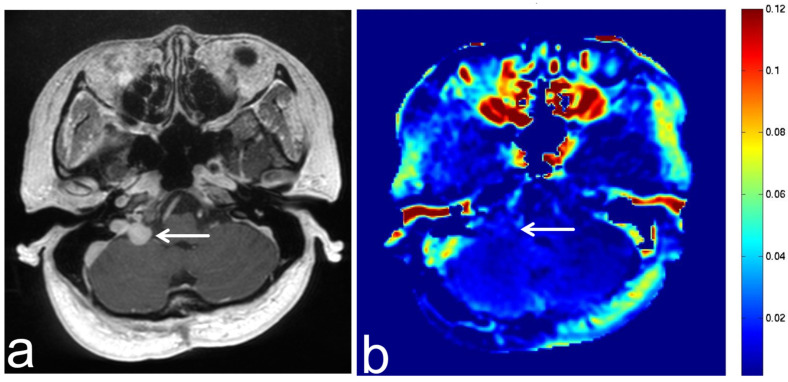
Images from a 46-year-old woman with a solitary fibrous tumor. (**a**) Three-dimensional, contrast-enhanced, T1-weighted sequence shows enhancement of a solitary fibrous tumor in the right cerebellopontine angle (white arrow). (**b**) APT-CEST sequence (axial section) shows low SI consistent with solitary fibrous tumor (white arrow).

**Figure 14 diagnostics-13-00914-f014:**
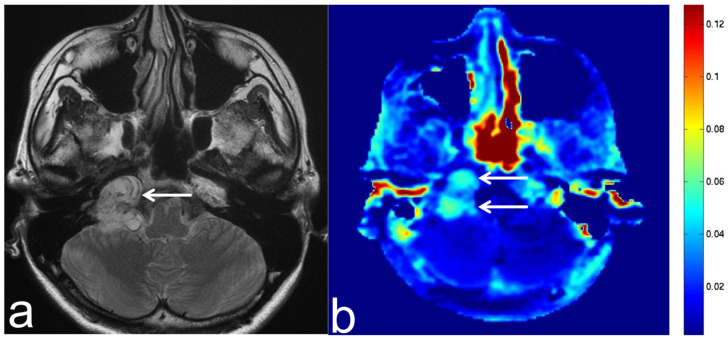
Images from a 38-year-old woman with a schwannoma. (**a**) T2-weighted sequence (axial section) shows a schwannoma in the right cerebellopontine angle with uneven high SI (white arrow). (**b**) APT-CEST sequence (axial section) shows high SI consistent with schwannoma (white arrow).

**Figure 15 diagnostics-13-00914-f015:**
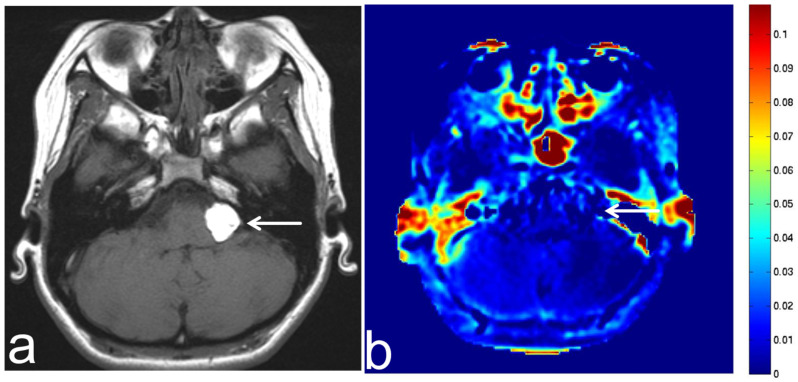
Images from a 24-year-old woman with a suspected lipoma. (**a**) T1-weighted sequence (axial section) shows a suspected lipoma in the left cerebellopontine angle with high SI (white arrow). (**b**) APT-CEST sequence (axial section) shows low SI consistent with suspected lipoma (white arrow).

**Figure 16 diagnostics-13-00914-f016:**
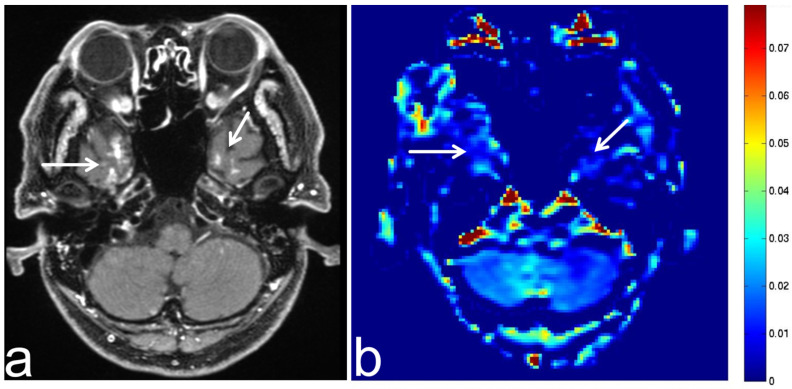
Images from a 52-year-old man with suspected radiation necrosis after irradiation for nasopharyngeal cancer. (**a**) Three-dimensional, contrast-enhanced, T1-weighted sequence shows multiple sparse enhancements in the bilateral temporal lobe (white arrow). Radiation necrosis is suspected according to the clinical course. (**b**) APT-CEST sequence (axial section) shows low SI consistent with the enhancement in the bilateral temporal lobe (white arrow).

**Figure 17 diagnostics-13-00914-f017:**
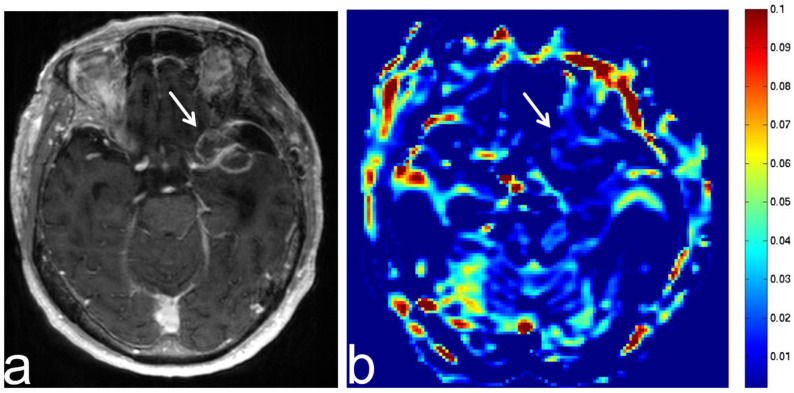
Images from a 70-year-old man with suspected radiation necrosis after operation and irradiation for brain metastasis from esophageal cancer. (**a**) Three-dimensional, contrast-enhanced, T1-weighted sequence shows multiple sparse enhancements in the left temporal lobe (white arrow). Radiation necrosis is suspected according to the clinical course. (**b**) APT-CEST sequence (axial section) shows low SI consistent with the enhancement in the left temporal lobe (white arrow).

**Figure 18 diagnostics-13-00914-f018:**
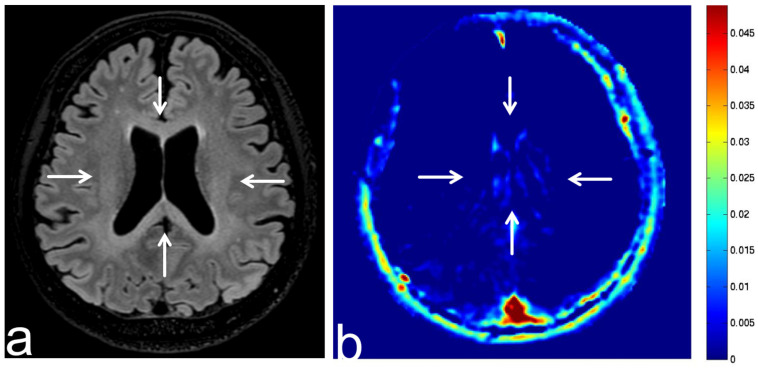
Images from a 50-year-old man with methotrexate-induced leukoencephalopathy. (**a**) Fluid-attenuated inversion recovery (FLAIR) sequence (axial section) shows low SI in periventricular white matter (white arrow). Leukoencephalopathy is suspected according to the clinical course. (**b**) APT-CEST sequence (axial section) shows low SI consistent with a leukoencephalopathy lesion (white arrow).

**Figure 19 diagnostics-13-00914-f019:**
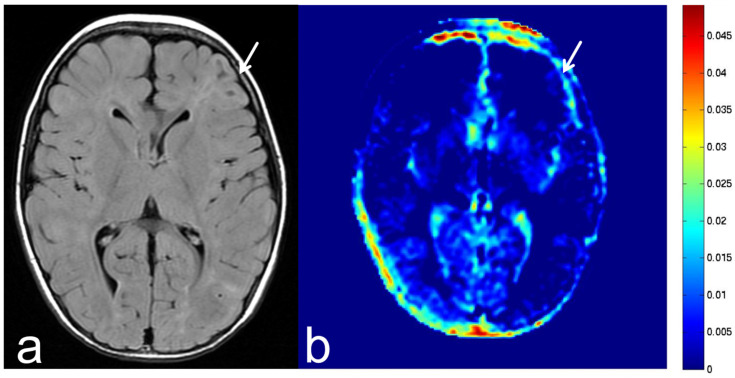
Images from a 4-year-old girl with tuberous sclerosis complex. (**a**) FLAIR sequence (axial section) shows slightly high SI in the left frontal lobe (white arrow). This lesion was suspected to be a cortical hamartoma. (**b**) APT-CEST sequence (axial section) shows low SI consistent with suspected cortical hamartoma (white arrow).

**Figure 20 diagnostics-13-00914-f020:**
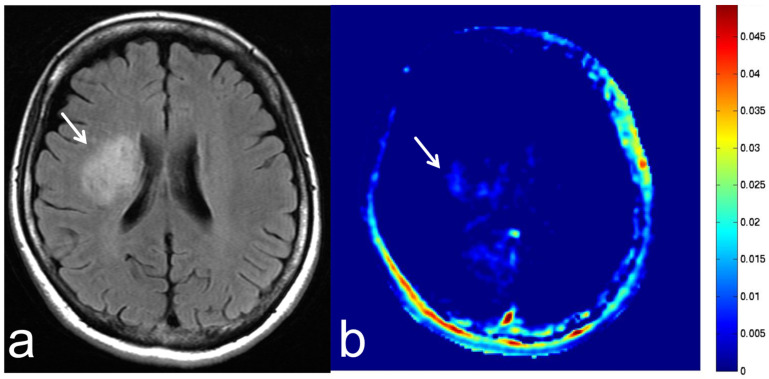
Images from a 43-year-old woman under treatment with an immunosuppressant drug for systemic lupus erythematosus who presented with progressive multifocal leukoencephalopathy (PML). (**a**) FLAIR sequence (axial section) shows PML with high SI in the right cerebral hemisphere (white arrow). (**b**) APT-CEST sequence (axial section) shows low SI consistent with PML (white arrow).

**Figure 21 diagnostics-13-00914-f021:**
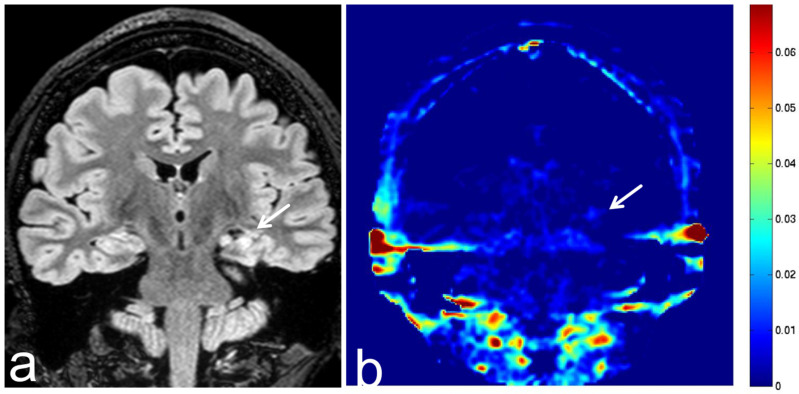
Images from a 36-year-old woman with hippocampal sclerosis. (**a**) FLAIR sequence (coronal section) shows left hippocampal sclerosis with high SI (white arrow). (**b**) APT-CEST sequence (coronal section) shows low SI consistent with hippocampal sclerosis (white arrow). There is no difference between the left and right hippocampal SI.

## Data Availability

The datasets during this study are not publicly available but are available from the corresponding author on reasonable request.

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
