# Peer review of "Amide Proton Transfer–Chemical Exchange Saturation Transfer Imaging of Intracranial Brain Tumors and Tumor-like Lesions: Our Experience and a Review"

_diagnostics, 2023, doi:10.3390/diagnostics13050914_

Round 1
Reviewer 1 Report
The authors present an interesting review about the role of amide proton transfer in neuroradiology, which is illustrated by interesting images. The subject falls within the scope of the journal. Description and discussion of the findings are well done and well-founded. The bibliography is pertinent and current. However, the text needs improvement. Excerpts that deserve special attention were marked in yellow and red.

Author Response
Response to Reviewer:
We are very grateful for your positive comments. We revised the manuscript while carefully considering your comments.
Chemical exchange saturation transfer (CEST) is a molecular-based magnetic resonance imaging (MRI) method that can generate image contrast based on the proton exchange between labeled protons in solutes and free bulk water protons. Amide proton transfer (APT) imaging is the most often reported amide proton-based CEST technique, generating image contrast reflecting the associations of mobile proteins and peptides resonating at 3.5 ppm downfield from water. Although the origin of the APT signal intensity in tumors is unclear, previous studies have suggested that APT signal intensity is increased in brain tumors because of increased mobile protein concentrations in malignant cells in association with increased cellularity. High grade tumors, which show higher proliferation than low grade tumors, have higher densities and numbers of cells (and higher concentrations of intracellular proteins and peptides) than low grade tumors, and APT-CEST imaging studies suggest that the APT-CEST signal intensity can be used to help differentiate between benign and malignant tumors? A low grade astrocytoma is not a benign tumor.
Response:
Thank you for your suggestion. I added a sentence (page 2, section 34-35).
and estimate the nature of lesions. In this review, we summarize the current applications and findings of the APT-CEST imaging of various brain tumors and tumor-like lesions. We report that APT-CEST imaging can provide additional information on intracranial brain tumors and tumor-like lesions over that provided by conventional MRI methods, and that it can help indicate the nature of lesions, differentiate between benign and malignant lesions, and determine therapeutic effects. Future research could initiate or improve lesion-specific clinical applicability of APT-CEST imaging for meningioma embolization, lipoma, leukoencephalopathy, tuberous sclerosis complex, progressive multifocal leukoencephalopathy, and hippocampal sclerosis.
Introduction Chemical exchange saturation transfer (CEST) is an important molecular-based magnetic resonance imaging (MRI) technique that can generate image contrast based on the proton exchange between labeled protons in solutes and free bulk water protons [1, 2]. Amide proton transfer (APT) imaging is the most often reported amide proton-based CEST technique, generating image contrast reflecting the associations of mobile proteins and peptides resonating at 3.5 ppm downfield from water [3, 4]. APT-CEST does not require exogenous contrast and is safe in patients with renal failure or with intolerance to contrast media. The technique has shown potential clinical utility for differentiating radiation necrosis from tumor recurrence or progression, and high-grade from low grade glioma [5-7]. Although the origin of the APT signal intensity (SI) in tumors is unclear, previous studies have suggested that the APT SI is increased in brain tumors because of increased mobile protein concentrations in malignant cells in association with increased cellularity [3, 6-11]. Previous reports reach a consensus view that APT SI is higher in highly malignant tumors than in less-malignant tumors, and also higher in malignant tumors than in non-tumor lesions such as radiation necrosis. However, there are few reports on the APT SI of benign tumors and tumor-like lesions including demyelinating disease. We believe that the APT-CEST technique is suitable for application in a wider ranging field, including benign tumors and tumor-like lesions such as demyelinating disease. Therefore, our aim was to review the current applications and findings of the APT-CEST imaging of various brain tumors and tumor-like lesions, to help differentiate between benign and malignant tumors and estimate the nature of lesions.
Mechanism of APT CEST imaging The APT effect is measured as a reduction in bulk water intensity due to chemical exchange of water protons with labeled backbone amide protons of endogenous mobile proteins and peptides in tissue [3, 4]. Thus, specific molecular information is obtained indirectly through the bulk water signal. Such labeling is accomplished using selective radiofrequency (RF) irradiation at the MR frequency of the backbone amide protons, which are 3.5 ppm downfield of the water resonance, causing saturation that is transferred to water protons (Figure 1).
APT-CEST imaging of malignant tumor 1. Oligodendroglioma Oligodendroglial tumors are diffuse infiltrating gliomas that most commonly arise in the frontal lobe, with a peak incidence in the fourth or fifth decade of life. They are the second most common intracranial glial tumors and are mainly composed of World Health Organization (WHO) grade II and III oligodendrogliomas [15, 16](Figure 2).
Figure 2. Images from a 35-year-old woman with an oligodendroglioma. (a) Three-dimensional 3D contrast-enhanced T1-weighted sequence shows ill-definition and faint enhancement of an oligodendroglioma in the left temporal lobe (white arrow). (b) APT-CEST sequence (axial section) shows wide low SI (white arrow) and parts with relatively high SI (red arrow), consistent with oligodendroglioma.
- Diffuse astrocytoma The 2016 WHO classification of diffuse astrocytic tumors indicates three grades with different aggressiveness [17]. Although diffuse astrocytoma (WHO grade II) is a relatively slow growing tumor with a median survival time of 5–8 years, it has a high recurrence rate due to diffuse infiltration of brain tissue and inherent malignant potential to transform into high-grade astrocytoma such as anaplastic astrocytoma and secondary glioblastoma [18, 19] (Figure 3).
Figure 3. Images from a 41-year-old man with diffuse astrocytoma post-operation. (a) Three-dimensional contrast-enhanced T1-weighted sequence shows ring enhancement in the right cerebellar hemisphere (white arrow).
Response:
Thank you for your comment. I changed a sentence (page 10, section 160).
Local recurrence or malignant transformation is suspected. (b) APT-CEST sequence (axial section) shows relatively high SI in the center of a ring of enhancement (white arrow).
- Glioblastoma
Glioblastoma (GBM) is the most common malignant brain tumor in adults. With a designation of WHO Grade IV, it is also the most lethal primary brain malignancy, with a median survival time of only 15 months [20, 21] (Figures 4, 5).
(a) Three-dimensional contrast-enhanced T1-weighted sequence shows ring enhancement of a GBM in the left in the left deep periventricular white matter and adjacent splenium of corpus callosum (white arrow).
Response:
Thank you for your comment. I changed a sentence (page 11, section 172-173).
(b) APT-CEST sequence (axial section) shows high SI consistent with GBM (white arrow) and extremely high SI consistent with a poor contrast area (red arrow).
Figure 5. Images from a 72-year-old woman with a GBM after operation and radiotherapy. (a) Three-dimensional contrast-enhanced T1-weighted sequence shows enhancement at the marginal dead space of the surgical site in the right temporal lobe (white arrow). Local recurrence is suspected according to the clinical course. (b) APT-CEST sequence (axial section) shows relatively high SI consistent with the enhancement (white arrow). APT-CEST imaging shows iso-intensity or mild punctate hyperintensity in low grade gliomas, which is significantly lower than that seen in high grade gliomas7 . There are also significant differences in APT SI between grade II and III glioma, and grade III and IV glioma [6]. In our case, APT SI was higher in GBM than in oligodendroglioma and diffuse astrocytoma.
- Malignant lymphoma
Intracranial lymphoma may present essentially as primary central nervous system (CNS) B-cell non-Hodgkin lymphoma (B-cell PCNSL), metastatic CNS lymphoma, intravascular lymphomatosis, T-cell PCNSL, and intracranial Hodgkin's lymphoma.
Insert the following references:
Slone HW, Blake JJ, Shah R, Guttikonda S, Bourekas EC. CT and MRI findings of intracranial lymphoma. AJR 2005;184:1679-1685.
Schwingel R, Reis F, Zanardi VA, Queiroz LS, França MC Jr. Central nervous system lymphoma: magnetic resonance imaging features at presentation. Arq Neuropsiquiatr. 2012 Feb;70(2):97-101.
Response:
Thank you for your suggestion. I added a sentence and references (page 13, section 193-196).
Primary central nervous system lymphoma is a rare form of non-Hodgkin lymphoma and isREMOVE responsible for 1%–2% of all central nervous system tumors. It encompasses lymphoma exclusively involving the brain, spinal cord, eyes, meninges, and cranial nerves, with 90%–95% classified histologically as diffuse large B-cell lymphomaREMOVE [22].
Response:
Thank you for your suggestion. I removed sentences (page 13, section 197-201).
Primary central nervous system lymphomas show more homogeneous APT SI than high grade gliomas. Maximum APT SI in primary central nervous system lymphomas was lower than in high grade gliomas [23]. In our case, B cell lymphoma showed high APT SI (Figure 6).
Figure 6. Images from a 68-year-old man with diffuse large B cell lymphoma. (a) Three-dimensional contrast-enhanced T1-weighted sequence shows an intense enhancement in an intraventricular mass malignant lymphoma (white arrow). (b) APT-CEST sequence (axial section) shows high SI throughout the intraventricular malignant lymphoma (white arrow).
Response:
Thank you for your comment. I added sentences (page 14, section 208-209).
- Brain metastasis Brain metastases (BMs) are associated with a poor prognosis irrespective of the primary tumor they originate from. Lung cancer, breast cancer, and melanoma are the most common causes of BM, and they account for 67%–80% of cases [24]. A previous report showed that APT SI in perilesional tissue in GBM were significantly lower in the solitary brain metastases [25]. Moreover, one report showed that the APT SI of solitary BMs was lower than that of enhancing areas of GBM [26]. In our case, the margin of BMs showed relatively high APT SI (Figure 7).
Figure 7. Images from a 70-year-old man with brain metastasis from esophageal cancer. Diagnostics 2021, 11, x FOR PEER REVIEW 8 of 28 (a) Three-dimensional contrast-enhanced T1-weighted sequence shows ring enhancement of brain metastasis in the left temporal lobe (white arrow). (b) APT-CEST sequence (axial section) shows relatively high SI at the margin of the brain metastasis (white arrow).
APT-CEST imaging of meningioma Meningioma is the most frequently diagnosed primary brain tumor in adults [27, 28]. Meningiomas are categorized into three WHO grades with 15 histological subtypes, indicating heterogenous clinical and molecular genetic characteristics [29]. Because most meningiomas are benign and categorized as WHO Grade I with a slow-growing behavior, most require no immediate treatment. However, some subtypes corresponding to WHO grades II and III are associated with a higher risk of recurrence and shorter survival times. One study investigated the feasibility of APT-CEST imaging for differentiating benign from atypical meningioma [30] (Figure 8), while another investigated the feasibility of APT-CEST imaging for differentiating growing meningiomas from non-growing meningiomas [31] (Figures 9-11). Preoperative transcatheter arterial embolization (TAE) of meningioma with polyvinyl alcohol microparticles has often been performed, and is considered a safe, efficient, and cost-effective method with few complications [32, 33]. However, to our knowledge, there are no reports of APT-CEST imaging in connection with embolization of meningioma. In our case, the APT SI of atypical meningioma decreased after TAE (Figure 8).
Figure 8. Images from a 64-year-old woman with an atypical meningioma before and after transcatheter arterial embolization (TAE). (a) Three-dimensional contrast-enhanced T1-weighted sequence shows atypical falcine meningioma involving the superior sagittal sinus before TAE (white arrow)
Response:
Thank you for your comment. I changed a sentence (page 17, section 250).
(b) Three-dimensional contrast-enhanced T1-weighted sequence shows atypical falcine meningioma involving the superior sagittal sinus after TAE (white arrow).
Response:
Thank you for your comment. I changed a sentence (page 17, section 252).
There was no apparent change in the atypical meningioma compared with pre-TAE imaging. (c) APT-CEST sequence (axial section) shows high SI consistent with meningioma before TAE (white arrow). (d) APT-CEST sequence (axial section) shows a decrease in SI in the left side of the atypical meningioma after TAE (white arrow). However, color bar and range for c and d were slightly different.
Figure 9. Images from a 74-year-old woman with a non-growing intracranial meningioma. (a) Three-dimensional contrast-enhanced T1-weighted sequence shows a convexity meningioma in the left temporal lobe (white arrow). (b) APT-CEST sequence (axial section) shows low SI consistent with meningioma (white arrow).
Figure 10. Images from a 77-year-old woman with a growing intracranial meningioma. (a) Three-dimensional contrast-enhanced T1-weighted sequence shows a dural attached lesion in the right posterior fossa (white arrow)
Response:
Thank you for your comment. I changed a sentence (page 19, section 270-272).
(b) APT-CEST sequence (axial section) shows relatively high SI consistent with meningioma (white arrow).
Figure 11. Images from a 67-year-old woman with a growing meningioma post operation. (a) (b) Three-dimensional contrast-enhanced T1-weighted sequence shows partially increasing cerebellopontine angle meningioma (white arrow). (c) APT-CEST sequence (axial section) shows relatively high SI consistent with the increasing part of the tumor (white arrow). APT-CEST imaging of benign lesion 1. Sphenoid sinus mucocele Paranasal sinus mucocele is defined as accumulation and retention of mucoid secretion within the sinus, leading to thinning, distension, and erosion of its bony walls. Sphenoid sinus mucocele is pathologically benign and comprises 1%–2% of all paranasal sinus mucoceles [34]. To the best of our knowledge, there are no reports of APT-CEST imaging of sphenoid sinus mucocele. However, this lesion is not a tumor, and low APT SI is expected. In our case, sphenoid sinus mucocele showed low APT SI (Figure 12).
Figure 12. Images from a 38-year-old man with sphenoid sinus mucocele. (a) T2-weighted sequence (coronal section) shows a sphenoid sinus mucocele (white arrow). (b) APT-CEST sequence (axial section) shows low SI consistent with sphenoid sinus mucocele (white arrow). Solitary fibrous tumor Solitary fibrous tumor (SFT) is a new combined entity for grade I–III soft-tissue tumors that was introduced in the 2016 World Health Organization classification of tumors of the central nervous system [17]. In our case, this tumor was pathologically diagnosed as grade I after surgery. There are no reports on APT-CEST imaging of SFT. However, this lesion is a benign tumor, and we expect that its APT SI will be lower than that of malignant tumor because SFT typically shows low proliferation. In our case, SFT showed low APT SI (Figure 13).
Figure 13. Images from a 46-year-old woman with a solitary fibrous tumor. (a) Three-dimensional contrast-enhanced T1-weighted sequence shows enhancement of a solitary fibrous tumor in the right cerebellopontine angle (white arrow). (b) APT-CEST sequence (axial section) shows low SI consistent with solitary fibrous tumor (white arrow).
- Schwannoma
Schwannoma is a benign tumor and develops from the Schwann sheath.
The cranial nerve that is most often affected (in 90% of cases) is the vestibulocochlear nerve (cranial nerve VIII), followed by the trigeminal nerve (cranial nerve V). Acoustic schwannomas grow into the cerebellopontine angle, displacing the brainstem and cerebellum. In most cases, these originate from within the internal auditory canal, the dilation of which is an early radiological sign of tumor growth.
Dalaqua M, do Nascimento FBP, Miura LK, Garcia MRT, Barbosa Junior AA, Reis F. Magnetic resonance imaging of the cranial nerves in infectious, neoplastic, and demyelinating diseases, as well as other inflammatory diseases: a pictorial essay. Radiol Bras. 2022 Jan-Feb;55(1):38-46.
Response:
Thank you for your recommendation. I changed and removed sentences (page 22,23, section 316-325).
This lesion is a benign tumor, and we expect that the APT SI will be lower than that of malignant tumor because the proliferation ability of schwannoma is expected to be low. However, schwannoma showed high APT SI in our case (Figure 14). Recent study showed that APT SI in schwannomas with high SI on T2 weighted sequence was higher than that of meningiomas [37].
Figure 14. Images from a 38-year-old woman with a schwannoma. (a) T2-weighted sequence (axial section) shows a schwannoma in the right cerebellopontine angle with uneven high SI (white arrow). (b) APT-CEST sequence (axial section) shows high SI consistent with schwannoma (white arrow). 4. Lipoma Intracranial lipomas are rare growths that represent less than 0.1% of all brain tumors. Most intracranial lipomas are asymptomatic and found incidentally on imaging while assessing other conditions [38-41]. There are no reports on the APT- CEST imaging of lipoma. This lesion is a benign tumor, and we expect the APT SI of intracranial lipoma to be lower than that of malignant tumors because the proliferation ability of lipoma is expected to be low. In our case, suspected lipoma showed low APT SI (Figure 15).
Figure 15. Images from a 24-year-old woman with a suspected lipoma. (a) T1-weighted sequence (axial section) shows a suspected lipoma in the left cerebellopontine angle with high SI (white arrow).
(b) APT-CEST sequence (axial section) shows low SI consistent with suspected lipoma (white arrow). APT-CEST imaging of demyelinating disease and tumor-like lesion 1. Radiation necrosis The occurrence and extent of radiation necrosis depends on the age of the patient receiving radiotherapy and the lesion volume [42]. A previous study reported that the APT SI of glioma was higher than that of radiation necrosis [5]. In our case, radiation necrosis showed low APT SI (Figures 16, 17).
Figure 16. Images from a 52-year-old man with suspected radiation necrosis after irradiation for nasopharyngeal cancer. (a) Three-dimensional contrast-enhanced T1-weighted sequence shows multiple sparse enhancements in the bilateral temporal lobe (white arrow). Radiation necrosis is suspected according to the clinical course. (b) APT-CEST sequence (axial section) shows low SI consistent with the enhancement in the bilateral temporal lobe (white arrow).
Figure 17. Images from a 70-year-old man with suspected radiation necrosis after operation and irradiation for brain metastasis from esophageal cancer. (a) Three-dimensional contrast-enhanced T1-weighted sequence shows multiple sparse enhancements in the left temporal lobe (white arrow). Radiation necrosis is suspected according to the clinical course. (b) APT-CEST sequence (axial section) shows low SI consistent with the enhancement in the left temporal lobe (white arrow).
- Leukoencephalopathy Toxic leukoencephalopathy is characterized by progressive damage to the white matter, and its causes include a wide spectrum of diseases that may injure and cause structural alteration to the white matter. The insults may be due to toxic metabolites secondary to chemotherapy or immunosuppressive therapy, environmental, or infectious in origin [43]. There are no reports on APT- CEST imaging of leukoencephalopathy. However, this lesion is not a tumor, and we expect the APT SI to be low because the proliferation ability of leukoencephalopathy is expected to be low. In our case, suspected lipoma???? showed low APT SI (Figure 18).
Response:
Thank you for your comment. We mistook and leukoencephalopathy was right (page 28, section 384).
Figure 18. Images from a 50-year-old man with methotrexate-induced leukoencephalopathy. (a) Fluid-attenuated inversion recovery (FLAIR) sequence (axial section) shows low SI in periventricular white matter (white arrow). Leukoencephalopathy is suspected according to the clinical course. (b) APT-CEST sequence (axial section) shows low SI consistent with a leukoencephalopathy lesion (white arrow).
- Tuberous sclerosis complex Tuberous sclerosis complex (TSC) is an autosomal dominant disorder with high clinical variability. It shows various features on brain imaging, including subependymal nodules, radial bands, cortical hamartomas, and subependymal giant cell astrocytomas [44].
Response:
Thank you for your comment. We added a word (page 29, section 398).
In our case, this lesion is suspected to be cortical hamartoma. There are no reports on APT-CEST imaging of TSC and hamartoma. The lesions are benign, and we expect that the APT SI will be lower than that of malignant tumor because the proliferation ability of hamartoma is expected to be low. In our case, suspected hamartoma showed low APT SI (Figure 19).
Figure 19. Images from a 4-year-old girl with tuberous sclerosis complex. (a) FLAIR sequence (axial section) shows slightly high SI in the left frontal lobe (white arrow). This lesion was suspected to be a cortical hamartoma. (b) APT-CEST sequence (axial section) shows low SI consistent with suspected cortical hamartoma (white arrow). 4. Progressive multifocal leukoencephalopathy Progressive multifocal leukoencephalopathy (PML) is a rare but often fatal brain disease caused by reactivation of the polyomavirus JC (JCV) [45]. PML almost exclusively affects immunocompromised patients such as those with HIV/AIDS [46].
In systemic lupus erythematosus , risk factors for CNS infection include disease activity, current and cumulative corticosteroid dose, and the use of other immunosuppressants.
INSERT THE FOLLOWING REFERENCES
Yang CD, Wang XD, Ye S, Gu YY, Bao CD, Wang Y, et al. Clinical features, prognostic and risk factors of central nervous system infections in patients with systemic lupus erythematosus. Clin Rheumatol 2007;26:895e901.
de Amorim JC, Torricelli AK, Frittoli RB, Lapa AT, Dertkigil SSJ, Reis F, Costallat LT, França Junior MC, Appenzeller S. Mimickers of neuropsychiatric manifestations in systemic lupus erythematosus. Best Pract Res Clin Rheumatol. 2018 Oct;32(5):623-639.
Response:
Thank you for your recommendation. I added a sentence and references (page 30, section 417-419).
There are no reports on APT-CEST imaging of PML. However, this lesion is not a tumor, and we expect the APT SI to be low because the proliferation ability of PML is expected to be low. In our case, PML showed low APT SI (Figure 20).
Figure 20. Images from a 43-year-old woman under treatment with an immunosuppressant drug for systemic lupus erythematosus who presented with progressive multifocal leukoencephalopathy (PML). (a) FLAIR sequence (axial section) shows PML with high SI in the right cerebral hemisphere (white arrow). (b) APT-CEST sequence (axial section) shows low SI consistent with PML (white arrow). 5. Hippocampal sclerosis Hippocampal sclerosis (HS) is the most common histopathologic abnormality found in adults with drug-resistant temporal lobe epilepsy [47]. There are no reports on APT-CEST imaging of HS. However, this lesion is not a tumor, and we expect APT SI to be low because the proliferation ability of HS is expected to be low. In our case, PML showed low APT SI (Figure 21).
Figure 21. Images from a 36-year-old woman with hippocampal sclerosis. (a) FLAIR sequence (coronal section) shows left hippocampal sclerosis with high SI (white arrow). (b) APT-CEST sequence (coronal section) shows low SI consistent with hippocampal sclerosis (white arrow). There is no difference between the left and right hippocampal SI.
Discussion Recent studies on glioma have demonstrated a positive correlation of APT SI with cell proliferation index [6, 8, 11]. These results suggest that high grade tumors, which show higher proliferation, have higher densities and numbers of cells (and higher concentrations of intracellular proteins and peptides) than low grade tumors.
Concentration of mobile proteins and peptides per cell may increase with the grade of glioma. In the study of Togao et al., tumors with necrosis showed higher APT SI than those without necrosis. Although it was difficult to confirm whether this was a direct relationship, highly concentrated mobile proteins and peptides in the extracellular space, such as microscopic necrosis or fluid collection in the microcystic space, might also increase APT SI in tumors. The alternation in tissue pH might affect APT SI.
van Zijl PC, Yadav NN. Chemical exchange saturation transfer (CEST): what is in a name and what isn't? Magn Reson Med. 2011;65(4):927–948
Zhou J, Payen JF, Wilson DA, Traystman RJ, van Zijl PC. Using the amide proton signals of intracellular proteins and peptides to detect pH effects in MRI. Nat Med. 2003;9(8):1085–1090.
Togao O, Yoshiura T, Keupp J, et al. Amide proton transfer imaging of adult diffuse gliomas: correlation with histopathological grades. Neuro Oncol. 2014;16(3):441-448.
Response:
Thank you for your recommendation. I added a sentence and references (page 33, section 453-459).
Our case also showed that high grade tumors tended to have higher APT SI than low grade tumors, and that demyelinating disease and tumor-like lesions tend to have low APT SI. TAE is a standard pre-operative procedure for meningiomas aimed at reducing intraoperative bleeding and facilitating dissection. In our case of atypical meningioma after TAE, contrast enhanced T1-weighted sequences show no apparent change to the meningioma, but APT-CEST imaging shows a decrease in SI (Figure 8). After the operation in our case, a part of the tumor showed necrosis and ischemic change on pathological analysis. APT-CEST imaging may be used as an indicator of therapeutic effects on tumors. Moreover, APT-CEST imaging does not involve injection of a contrast agent and can be used in patients with renal failure and those who show adverse reactions to contrast media.
There have been few reports on APT-CEST imaging of benign lesions. In our case, they tended to show low APT SI. However, schwannoma (Figure 14) shows high APT SI consistent with T2 hyperintensity. The area showing T2 hyperintensity suggests a cystic or heterogeneous appearance. Small schwannomas are usually homogeneous and are histologically composed of Antoni type A pattern, while heterogeneous and cystic schwannomas are larger and include Antoni B pattern or a mix of type A and B patterns [48]. Therefore, this area may show higher proliferation and high APT SI, and one study showed higher APT SI in schwannomas with high SI on T2 weighted sequence compared to that of meningiomas [37].
Zhang et al. indicate that APT imaging may have some value in the determination of malignant brain tumour boundaries, but there is no doubt that this approach can be used in the differential diagnosis between gliomas and meningiomas.
Zhang HW, Liu XL, Zhang HB, et al. Differentiation of Meningiomas and Gliomas by Amide Proton Transfer Imaging: A Preliminary Study of Brain Tumour Infiltration. Front Oncol. 2022;12:886968.
Response:
Thank you for your recommendation. I added a sentence and references (page 34, section 482-484).
Similarly, the non-contrast region of GBM (Figure 4) also shows high APT SI, which suggests liquid containing high concentrations of intracellular proteins and peptides. However, sinus mucocele (Figure 12) shows low APT SI, because although this lesion may contain viscous fluid, there are less intracellular proteins and peptides. However, the number of cases examined is small, and further studies are needed. In addition, because the evaluations described in this study were made based on visual assessment, quantitative evaluation is also necessary. Conclusions APT-CEST imaging can provide additional information on intracranial brain tumors and tumor-like lesions over that provided by conventional MRI methods, which may help indicate the nature of lesions, differentiate between benign and malignant lesions, and determine therapeutic effects. Future research could initiate or improve lesion-specific clinical applicability of APT-CEST imaging for meningioma embolization, lipoma, leukoencephalopathy, TSC, PML, and HS. For these purposes, we may have to answer specific research questions and suggest guidelines to allow the development of APT-CEST imaging to reach its full potential.

Reviewer 2 Report
It is an interesting manuscripts review,
but it still is not a quantitative result.
The MRI imaging processing and non-parametric statistics analysis could be done for this review paper.
Amide Proton Transfer-Chemical Exchange Saturation Transfer imaging of intracranial brain tumors and tumor-like lesions: a review
A review of applications and findings of the APT-CEST imaging of various brain tumors and tumor-like lesions, between benign and malignant tumors is discussed in this manuscript.
It is an interesting manuscripts review, but it still is not a quantitative result
1:
There is a small confuse about the title word “review” due to in manuscript “in our case” is said. Is it a review work?
2:
In last paragraph of the discussion section is said
"However, the number of cases examined is small, and further studies are needed. In addition, because the evaluations described in this study were made based on VISUAL assessment, quantitative evaluation is also necessary."
I agree with author comments, in your opinion, how to perform a quantitative description of the APT-CEST higher effect? but included only the reported cases in this manuscript
It is viable for you to perform an imaging processing for these cases?
3:
I was wondering if this MRI modality would help in an more early diagnostic, such the doctor may hope a better recuperation in the patients.

Author Response
Response to Reviewer:
We are very grateful for your positive comments. We revised the manuscript while carefully considering your comments.
A review of applications and findings of the APT-CEST imaging of various brain tumors and tumor-like lesions, between benign and malignant tumors is discussed in this manuscript.
It is an interesting manuscripts review, but it still is not a quantitative result
1:
There is a small confuse about the title word “review” due to in manuscript “in our case” is said. Is it a review work?
Response:
Thank you for your question. As you say, it's not an only review. it's a report about our facility's experience and previous literature.
Therefore, we changed “Amide Proton Transfer-Chemical Exchange Saturation Transfer imaging of intracranial brain tumors and tumor-like lesions: our experience and a review”
2:
In last paragraph of the discussion section is said
"However, the number of cases examined is small, and further studies are needed. In addition, because the evaluations described in this study were made based on VISUAL assessment, quantitative evaluation is also necessary."
I agree with author comments, in your opinion, how to perform a quantitative description of the APT-CEST higher effect? but included only the reported cases in this manuscript
It is viable for you to perform an imaging processing for these cases?
Response:
Thank you for your question. We can evaluate APT-CEST effect of brain lesions by placing ROI after APT-CEST image is taken. However, it is difficult to evaluate the entire lesion because only one cross section of the lesion can be evaluated.
3:
I was wondering if this MRI modality would help in an more early diagnostic, such the doctor may hope a better recuperation in the patients.
Response:
Thank you for your comment. Since this MRI modality does not use a contrast agent, it is useful to infer the properties just by adding it to the plain MRI. As a result, we believe that it may be useful for early diagnosis.
